# Induced CAR-Macrophages as a Novel Therapeutic Cell Type for Cancer Immune Cell Therapies

**DOI:** 10.3390/cells11101652

**Published:** 2022-05-16

**Authors:** Siyu Su, Anhua Lei, Xudong Wang, Hengxing Lu, Shuhang Wang, Yuqi Yang, Ning Li, Yi Zhang, Jin Zhang

**Affiliations:** 1Quanzhou First Hospital Affiliated to Fujian Medical University, Quanzhou 362000, China; susiyu0312@163.com; 2Zhejiang Laboratory for Systems and Precision Medicine, Zhejiang University Medical Center, 1369 West Wenyi Road, Hangzhou 311121, China; leiah@nicemice.cn (A.L.); 21518452@zju.edu.cn (X.W.); hengxingl@126.com (H.L.); 3Center for Stem Cell and Regenerative Medicine, Department of Basic Medical Sciences, The First Affiliated Hospital, Zhejiang University School of Medicine, Hangzhou 310058, China; 4Institute of Hematology, Zhejiang University, Hangzhou 310058, China; 5National Cancer Center/National Clinical Research Center for Cancer, Cancer Hospital, Chinese Academy of Medical Sciences and Peking Union Medical College, 17 Panjiayuan, Beijing 100021, China; snowflake201@gmail.com (S.W.); lining@cicams.ac.cn (N.L.); 6NHC Key Laboratory of Pulmonary Immunological Diseases, Guizhou Provincial People’s Hospital, No. 83 Zhongshan Road, Guiyang 550000, China; 18520224337@163.com

**Keywords:** chimeric antigen receptor (CAR), human primary macrophage cells, induced pluripotent stem cells (iPSC)-derived macrophage cells (iMac), anticancer cell functions

## Abstract

The Chimeric antigen receptor (CAR)-T cell therapy has made inroads in treating hematological malignancies. Nonetheless, there are still multiple hurdles in CAR-T cell therapy for solid tumors. Primary CAR-expressing macrophage cells (CAR-Ms) and induced pluripotent stem cells (iPSCs)-derived CAR-expressing macrophage cells (CAR-iMacs) have emerged as attractive alternatives in our quest for an efficient and inexpensive approach for tumor immune cell therapy. In this review, we list the current state of development of human CAR-macrophages and provide an overview of the crucial functions of human CAR-macrophages in the field of tumor immune cell therapy.

## 1. Introduction

In recent years, there has been a burgeoning interest in harnessing immune cell therapy for solid cancers. Inspired by the pattern that T-cell antigen receptor (TCR)-formed immunological synapse stimulates immune activation of cytotoxic T cell, chimeric antigen receptor (CAR) has emerged in response to the desire to use engineered cell products for much more effective treatment. CAR is a type of synthetic chimera protein which encompasses an extracellular antigen-recognition domain, a transmembrane domain, and an intracellular domain [1]. The extracellular domain consists of a single-chain variable fragment (scFv) responsible for antigen recognition and a hinge region (Hinge). The intracellular domain comprises a costimulatory domain and a signal transduction domain. Therefore, the CAR has been designed to improve the targeting efficacy of immune cells and induce cell antigen-dependent activation. Recently, the chimeric antigen receptor (CAR)-T cell therapy has achieved significant success in treating hematologic malignancies [2,3,4]. Nevertheless, there are still obstacles in the application of CAR-T cell therapy against solid cancer, such as off-target effects, poor efficiency against infiltrating tumor lumps and CAR-T cell exhaustion resulting from immunosuppressive tumor microenvironment (TME), as well as the loss or decrease of antigens in solid tumor cells [5,6]. Moreover, the high costs associated with CAR-T cell procedures restricted the widespread application of CAR-T in clinical treatment [3]. Consequently, much emphasis has been placed on developing new alternatives, including optimizing chimera antigen receptors, improving T-cell capacity, exploiting the different features of subsets of T cell or NK cells, and making the so-called off-the-shelf universal cells. Despite these significant attempts, efforts to optimize CAR-T cells still have a long way to go.

Macrophages are well established for their unique roles in immunoregulation and capacity to infiltrate solid tumors [7,8]. Traditionally, macrophages have been defined as highly plastic cells which serve plenty of functions such as elimination of pathogens, clearance of cellular debris, regulation of inflammatory responses, and contributing to tissue development and homeostasis [9]. All the studies suggest that macrophages will be a new choice for tumor immune cell therapy. However, research indicated that, in the particular microenvironment, macrophages could stay in two polarized states in the presence of diverse cytokines and antigens, i.e., classically activated (antitumor and antibacteria) macrophages (M1) and alternatively activated (tissue remodeling and immunomodulatory properties) macrophages (M2) [10,11,12]. Consequently, efforts have been made to reverse M2 macrophages into the M1 phenotype of macrophages or adoptively transplant genetically engineering CAR-macrophages into patients. In this review, we introduce the influence of macrophages on the tumor microenvironment and provide an overview of the three main sources of human CAR-macrophages and their role in eliminating cancer cells to provide a foothold for future works.

## 2. Crosstalk between Macrophages and Tumor Microenvironment

Current evidence suggests that tumor-associated macrophages (TAMs) account for the majority of infiltrating immune cells in tumor tissue [13,14] and orchestrate the formation of the immunosuppressive environment by recruiting T regulatory cells (**Tregs**) and Myeloid-derived suppressor cells (MDSCs), secreting inhibitory cytokines, inducing hyper-expression of PD-L1, which is the ligand of programmed cell death protein 1 (PD-1) of T cells and altering the metabolic profile of immune cells [13,15]. It has been shown that TAMs cooperate with MDSCs by secreting CCL20 in a mouse colorectal cancer (CRC) model [16] and recruit Tregs by producing CCL22 in human ovarian carcinomas to establish the immunosuppressive TME [17]. Moreover, peroxynitrites produced by TAMs can prevent the recruitment of T cells by mediating the nitration of CCL2 and CCL5, thus protecting tumor cells from immune attack [13,18]. Overwhelming evidence substantiates that TAMs-derived immune suppressive factors such as IL-10, CXCL10, TGF-β, and ROS restrain T lymphocytes from eliminating cancer cells [19,20,21,22,23]. On the other hand, cytokines TNF-α and IL-6 secreted by infiltrated macrophages reportedly promote the expression of PD-L1 via the Nuclear Factor-kappa B (NF-kB) and STAT3 signaling pathways in gastric cancer cells [24]. Meanwhile, another recent work indicated that TAMs promoted PD-L1 expression in pancreatic ductal adenocarcinoma (PDAC) via TGF-β1-induced nuclear translocation of PKM2/STAT1 complex [25]. Consequently, upregulation of PD-L1 in macrophages induced not only T cells exhaustion by binding to PD-1, but also M2 polarization of macrophages attended by mitochondrial function improvement and metabolic reprogramming via Erk/Akt/mTOR [26]. Apart from PD-L1, PD-1 expression has also been identified in both mouse and human TAM and promotes an M2-like profile which contributes negatively to phagocytosis against tumor cells [27]. Otherwise, the unique metabolism of TAMs also directly suppresses the effector functions of T cells through modulating changes in metabolic profile. For example, acetyl-CoA and glycine generated methylglyoxal in TAMs contribute to T-cell dysfunction by depleting intracellular enzyme resources in T cells by generating methylglyoxal-derived argpyrimidine and hydroimidazolone [28,29]. In addition to being regarded as drivers of a tumor-promoting related anti-inflammation environment, macrophages directly bolster tumor progression by stimulating cancer stem cell proliferation via WNT signaling and promoting angiogenesis via production of VEGF, paving the way to tumor cell metastasis and invasion via remodeling of the extracellular matrix (ECM) and paracrine activities of epidermal growth factor (EGF) family ligands [13,30,31,32,33].

Given the critical promoting role of M2 macrophages in the progression of solid tumors, reprogramming M2 macrophages into a pro-inflammatory phenotype in the TME represents a promising therapeutic approach. To achieve this goal, radiotherapy via fractionated cumulative radiation dose regimens [34], chemotherapy approaches such as histone deacetylases inhibitor and chemical agents against CSF1 and CXCR4 [35,36,37], and immunotherapy with antibodies including anti-CD40 anti-CD47 have been utilized to target TAMs in cancer therapy [38,39,40,41]. Since the better prognosis associated with M1 macrophages in cancer patients has been validated in pre-clinical and clinical research [42,43], the efforts to reverse the M2 state of innate macrophages into the M1 phenotype would play a role to some extent. Nonetheless, these attempts have had little lasting effect. Further studies are warranted to obtain more effective treatment to enhance the activity of macrophages for tumor treatment, avoid the transformation of TAMs to M2 macrophages and boost the killing of tumor cells by M1 macrophages. Recent studies have uncovered a new type of immune cell, genetically engineered human macrophage with CARs, providing a novel choice for immunotherapy of neoplasms [44,45].

## 3. Sources of Human CAR-Macrophages Production

Macrophages are well recognized as effector cells that eliminate cancer cells due to their phagocytic ability, and are therefore drawing attention in the field of immunotherapy of cancers [46,47]. In this section, we compare different sources of human macrophages and the corresponding methods of CAR transgene delivery into macrophages.

### 3.1. Human THP-1 Macrophage Cell Line

THP-1, a human leukemia monocytic cell line isolated from a one-year-old male patient, was found to share some canonical morphological and functional properties with primary monocytes and macrophages and the expression of differentiation markers [48,49]. Methods have been developed to differentiate THP-1 into macrophage-like cells using phorbol-12-myristate-13-acetate (PMA) [50,51,52], 25-dihydroxyvitamin D3 (vD3) [53] or macrophage colony-stimulating factor (M-CSF) [54] (Figure 1A). THP-1 macrophages express typical myeloid cell surface markers, i.e., CD11b, CD36, TNF-α, CCR7, TLR-2, etc., and exhibit macrophage morphological characteristics and phagocytosis potency. Interestingly, the M1 phenotype can be observed when stimulating THP-1 macrophages with lipopolysaccharide (LPS) and IFN-γ, similar to the pro-inflammatory polarization in primary macrophages [55,56]. Furthermore, THP-1 monocytes have a homogeneous genetic background, are relatively easy to culture, proliferate fast, and can be easily stored, making it relatively easy and safe to differentiate into macrophages [57,58,59,60]. Accordingly, THP-1 has been harnessed to reveal modulation of monocyte/macrophage physiological behavior, such as immune response, cell–cell interaction, and signaling pathway transduction [61]. Moreover, researchers have attempted to assess the antigen-dependent phagocytosis capability of THP-1-derived macrophages. The first-generation anti-CD19 CAR encoding the CD3ζ intracellular domain has been transduced into THP-1 cells (Figure 1A). Intriguingly, CD3ζ shares homology with the Fc common γ-chain (FcεRI-γ), which is a specific classical signaling molecule in macrophages participating in antibody-dependent cellular phagocytosis (ADCP). Besides, CD3ζ-based CARs could enhance the antitumor phagocytic activity of THP-1 macrophages [45]. This work provided a foothold for subsequent studies aiming to translate this platform to primary human macrophages. However, despite the close similarity of gene expression profiles after LPS stimulation between THP1 and PBMC-derived macrophages, it remains unclear to what extent the THP-1 macrophages can mimic primary macrophages. For example, LPS-induced tolerance usually observed in primary macrophages can be dismissed in THP-1 macrophages because of the upregulation of specific genes such as NF-κB, causing persistent sensitivity to LPS [62,63,64,65]. Moreover, during M1 polarization, the absence of expression of IL-6 and IL-10 and lower secretion of IL-8 in THP-1 make it different from human primary monocytes derived from peripheral blood monocytic cells (PBMC) [66].

### 3.2. Human Macrophages from PBMC

PBMC purified from fresh blood consists of multiple types of immune cells, including monocytes/macrophages, making it an ideal material for exploring the function of innate immune cells. Traditionally, compared with THP-1 monocytes/macrophages, PBMC-derived monocytes/macrophages retain their primary morphology [67]. Upon M1 stimulation, PBMC-derived primary monocytes/macrophages can reportedly produce greater amounts of pro-inflammatory factors, e.g., IL-6, IL-8, TNF-α, etc. [66], and higher levels of surface markers, including natriuretic peptide receptor (NPR), CD14, CD68, indicating more robust inflammatory properties compared to THP-1 [68]. Meanwhile, researchers also attempted to generate primary human macrophages from peripheral blood CD14^+^ monocytes by treating them with GM-CSF (Figure 1B). However, a limitation for primary human macrophages is the difficulty of genetic manipulation. To overcome this obstacle, a replication-incompetent chimeric adenoviral vector (Ad5F35) has been designed to transduce CAR into the macrophages to harvest CAR-Ms. This strategy showed high efficiency and reproducibility among various donors [45,69]. After validating this approach of CAR delivery into macrophages, CAR-Ms targeting the solid tumor antigens Mesothelin or HER2 were generated (Figure 1B). Nevertheless, the low yield of macrophages from PBMC still impedes its large-scale application in clinics.

### 3.3. Pluripotent Stem Cell-Derived Car-Macrophage Cells

Despite exhibiting significant advantages in immunotherapy against solid cancers, conventional macrophages have demonstrated limitations when applied during patient treatment. In this regard, immortalized macrophage cell lines are not feasible for clinical applications, and bone marrow or PBMC-derived primary macrophages are not efficient enough to be engineered. In recent years, significant efforts have been undertaken to develop new approaches to harvest robust, genetic manipulation-friendly, and amount-plentiful macrophages. To bridge this knowledge gap, Zhang et al. [70] developed a novel strategy by inducing iPSCs into CAR-expressing macrophages (CAR-iMacs) equipped with innate immune functions, such as expression and secretion of immune-related cytokines, repolarization of M2 state toward the pro-inflammatory M1 phenotype in an antigen-dependent way, as well as phagocytosis against aberrant cells, and antitumor capacity [70]. This method has been considered the first attempt to eliminate cancer cells via source-unlimited genetically engineered CAR-iMacs to solve the challenges associated with primary immune cell therapies.

Specifically, PBMCs isolated from healthy donors were reprogrammed by five pluripotency factors into iPSC (Figure 1C). After obtaining single iPSC clones, CD3ζ-contained CAR selected from different CD19-targeted CARs were introduced into the iPSC clone cells to obtain CAR-iPSCs through lentiviral transduction (Figure 1C). Subsequently, the CAR-iPSCs were induced to differentiate toward myeloid cell lineages mimicking the physiological development process of myeloid/macrophage differentiation to acquire nearly 100% CD14 and CD11b positive induced macrophage cells. Of note, the genetically engineered and induced products yielded by this method exhibited quintessential macrophage marker gene expression and signatures, confirmed by a multitude of experiments, such as RNA-sequencing, GO analyses, single-cell RNA-sequencing analysis, and trajectory analysis [70]. As expected, accompanied by the upregulation of crucial macrophage marker genes, pluripotent marker genes tended to completely disappear early in the differentiation process. Consequently, iPSCs-induced CAR-iMacs possess macrophage characteristics and show promising potential in tumor immunotherapy.

## 4. Applications Assessment

In this section, we report current studies that have validated the antigen-dependent activation potency of CAR-macrophages in vitro. Furthermore, we listed several in vivo experimental results that validated the antitumor capacity of CAR-macrophages in an antigen-dependent way and corroborated the clinical application value of CAR-macrophages.

### 4.1. Human Macrophage THP-1 Cells and Human Macrophages from PBMC-Derived CAR-Ms

#### 4.1.1. In Vitro Experiments

Further assays have been designed to confirm the antigen-specific phagocytosis of anti-HER2 CAR-Ms against HER2^+^ beads and tumor cells. When designed to eliminate SKOV3, a HER2^+^ ovarian cancer cell line, CAR-Ms exerted the cytotoxicity function in a dose- and time-dependent manner [45]. What is more, the level of CAR expression and the level of corresponding antigen exposure could affect the tumor-cell-killing ability of CAR-Ms [45].

Next, experiments to validate the phenotypic plasticity of CAR-Ms in vitro showed that untransduced (UTD) macrophages instead of CAR-Ms could upregulate the M2 marker CD206 when stimulated with IL4-, IL13-, or SKOV3-conditioned media. Accordingly, UTD macrophages stimulated with IL4 showed an evident increase in basal oxygen consumption rate, indicating the M2 polarization tendency of the wild-type macrophages, while CAR-Ms could withstand remodeling towards the M2 phenotype. Moreover, transcriptome analysis indicated that IL4 or IL13 triggered fewer genes, including MRC1, in CAR-Ms than UTD macrophages [45].

Furthermore, in vitro experiments were conducted to verify the effect of CAR on surrounding macrophages which are likely to exist in an immunosuppressive TME, and found that CAR-Ms induced the activation of macrophage-specific pro-inflammatory pathways, such as interferon signaling and iNOS signaling pathways. Moreover, CAR-Ms-conditioned media could lead to a phenotypic shift in wild-type macrophages towards the pro-inflammatory state (M1). Finally, exposure to CAR-Ms resulted in the expression and localization of the pro-inflammatory markers on the macrophage plasma membrane. Overall, CAR-macrophages strongly affected surrounding immune cells, which accounted for CAR-Ms’ antitumor activity to a certain extent [45].

Based on the interaction between macrophages and lymphocytes, experiments were performed to assess the stimulation and antigen-presenting capability of CAR-Ms to T cells. The above findings suggested that the combination of CAR-Ms and the tumor-associated antigen NY-ESO1-expressing tumor cells led to more robust activation of anti-NY-ESO-1 CD8^+^ T cells than the CAR-Ms alone or tumor cells alone.

#### 4.1.2. In Vivo Experiments

Two procedures have been conducted to explore CAR-Ms’ anticancer effects in NOD-scid IL2Rg_null_-3/GM/SF (NSGS) mice. The first approach involved intravenous (IV) injection of SKOV3 cells to trigger lung metastases and then dividing mice into different groups that received various treatments via a single IV injection. The other approach was to model intraperitoneal (IP) carcinomatosis using a similar approach to offer distinctive management with a single IP injection. It was concluded that CAR-Ms resulted in significant tumor deterioration, while UTD macrophages did not [45]. In addition, the treatment significantly enhanced overall survival, and was not associated with any obvious toxicity or body weight changes.

Importantly, persistence, biodistribution, and even chemotaxis ability of adoptive transplanted immune cells in living bodies have been associated with antitumor efficacy. The authors reported that CAR-Ms could survive at least 62 days in tumor-free NSGS mice. Furthermore, adenovirus-based genetic manipulation did not destroy the expression profile of chemokine receptors, meaning intact migrating and infiltrating ability of CAR-Ms towards relevant chemokines. Interestingly, CAR-Ms accumulated in the liver, regarded as an innate immune organ after IV injection in all experimental mice.

Subsequently, given that adenoviral infection can activate the inflammasome [71,72], it was hypothesized that exposure to Ad5F35 might lead to a pro-inflammatory (M1) macrophage phenotype. This evidence was obtained from several experiments, such as transcriptomes analysis, quantitative PCR, and flow cytometry, which supported their hypothesis, and the M1 phenotype remained for at least 40 days after transduction [45].

Moreover, the dynamic interaction of CAR-Ms with the human tumor microenvironment (TME) in a humanized immune system (HIS) mouse model was next elucidated. HIS mouse models generated through the transplantation of HSCs in NSG mice were treated with CAR-Ms. The single-cell RNA-Seq analysis showed that CAR-Ms and UTD macrophages isolated from HIS mice shared distinct phenotypes within the humanized TME, and CAR-Ms maintained pro-inflammatory features compared to UTD macrophages.

Finally, similar to in vitro studies, an experiment in vivo was performed to assess the potential of CAR-Ms to stimulate T cells. As expected, tumor-bearing NSGS mice that received combination therapy of CAR-Ms and donor-derived T cells displayed a favorable prognosis compared to the mice which received treatment of the single type of CAR-Ms or T cells. This finding illustrated that the interplay between adoptive-transplanted CAR-Ms and endogenous T cells should be taken into the evaluation of syngeneic immunocompetent murine tumor models in future studies.

### 4.2. Pluripotent Stem Cell-Derived CAR-Macrophage Cell: CAR-IMac

#### 4.2.1. In Vitro Experiment

Zhang et al. successfully obtained CAR (CD19)-iMacs and CAR (Mesothelin)-iMacs cells using the iPSC-derived CAR-iMacs approach and subsequently examined their antitumor potency by incubating with CD19-expressing K562 leukemia cells or Mesothelin-expressing OVCAR3/ASPC1 ovarian/pancreatic cancer cells, respectively, in vitro [70]. Based on the experimental results, they concluded that CAR-iMacs showed antigen-dependent enhanced phagocytosis activity against tumor cells. Upon recognition of antigens, CAR conferred antigen-dependent activation of intracellular signaling and promoted phosphorylation of ERK and NF-κB(P65) proteins, as well as expression of pro-inflammatory factors. Finally, the transcriptional analysis showed that, compared with WT-iMacs, CAR-iMacs possessed more M1 features in the TME.

#### 4.2.2. In Vivo Experiment

Similarly, when injected into NSG mice, the CAR-iMacs proliferated in vivo for 3 days; the therapeutic effect persisted for 3–4 weeks and gradually disappeared [70]. Furthermore, the researchers evaluated the antitumor cell activity by intraperitoneally injecting ovarian cancer cells HO8910 expressing a luciferase gene into NSG mice, followed by CAR-iMacs pretreated with IFN-γ to polarize them towards M1. The CAR-iMac-treated mice showed less tumor burden compared to the control group.

## 5. Advantages and Challenges of CAR-Macrophages Immunotherapy

Despite the substantial progress CAR-T cell therapy has achieved in clinical treatment [73], many challenges remain, such as cytokine release syndrome (CRS), CAR neurotoxicity, and drug resistance [74,75]. On the other hand, following T cells and NK cells, CAR-edited macrophages have exhibited promising potency against tumor cells in the past couple of years [45,47,70].

Compared with CAR-T cells, CAR-macrophages bring several unique benefits. T cells exhibit limited ability to reach the tumor environment due to the matrix-forming physical barriers throughout the tumor cells, while macrophages can effortlessly submerge in the cancer environment [76]. Additionally, CAR-macrophages can decrease the ratio of TAMs and switch the phenotype of TAMs. It should be noted that TAMs have significant functions in tumor infiltration, immunosuppression, and metastasis [77]. Moreover, CAR-macrophages play a phagocytic role in tumor cells, facilitate antigen presentation and augment the cytotoxicity of T cells. Finally, CAR-macrophages have less toxicity and limited circulation time than T cells [74].

Nonetheless, though macrophages have been studied in vivo and in vitro for many years, their genetic modification to improve their application to solid tumor therapy remains challenging. Further studies are required to demonstrate the effectiveness of CAR-macrophages in immunotherapy. Given that macrophages cannot proliferate by themselves in vitro or in vivo, their limited cell number is a significant challenge in CAR-macrophages immunotherapy. Moreover, little is known about the dose of macrophages required in vivo. Exogenous macrophages cross the lung after injection and predominantly linger in the liver [78], lowering treatment efficacy. Finally, it should be borne in mind that the TME is highly intricate, warranting further studies to improve current understanding [79].

## 6. Conclusions

Remodeling TAMs is an exciting and promising strategy for generating antitumor effects during the immunotherapy of neoplasms. Among the multiple efforts, adoptive transfer of genetically modified macrophages represents a promising therapeutic prospect. Accordingly, macrophages equipped with tumor antigen-specific CARs exhibit significantly improved cytotoxicity against tumor antigen-expressing cells and potentially reprogram the surrounding microenvironment [45] (Figure 2). Except for the direct antitumor activity, the CAR-macrophages exhibited an indirect therapy effect by stimulating the anti-neoplasm potency of tumor-infiltrated T cells. Compared with the former, Zhang et al. [70] effectively eliminated tumor cells in tumor-bearing mice by first providing an unlimited source of iPSC-derived engineered CAR-iMacs. However, despite the achievements made, the efficacy of CAR-iMacs treatment warrants further improvement by designing more potent CARs or further cell modifications.

## Figures and Tables

**Figure 1 cells-11-01652-f001:**
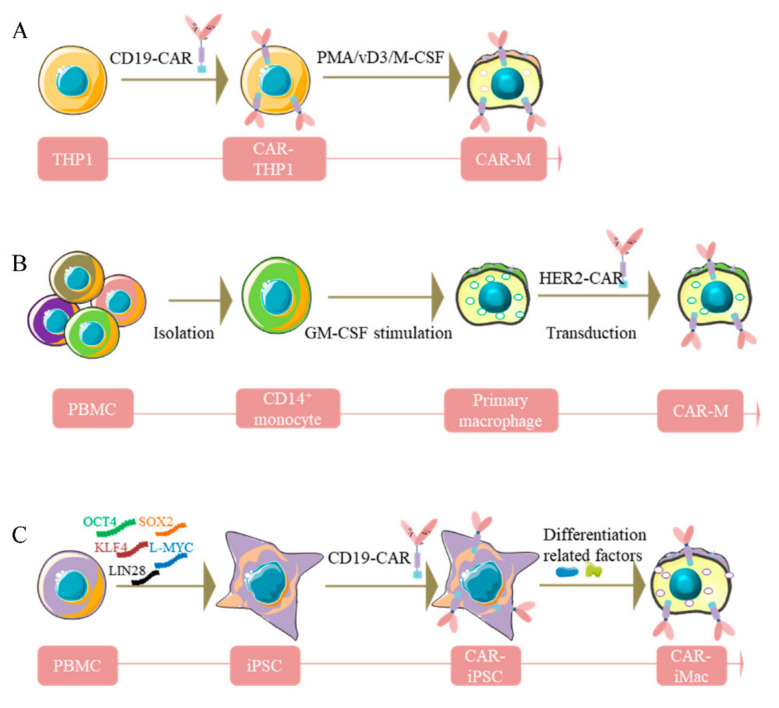
Diagram of the generation of CAR-macrophages. (**A**), Human THP-1 cells were engineered with CAR and then differentiated into CAR-macrophages. (**B**), Primary human macrophages were generated from peripheral blood CD14^+^ monocytes with GM-CSF and transduced to express CAR. (**C**), iPSCs were reprogrammed from PBMC, then differentiated into CAR-iMacs.

**Figure 2 cells-11-01652-f002:**
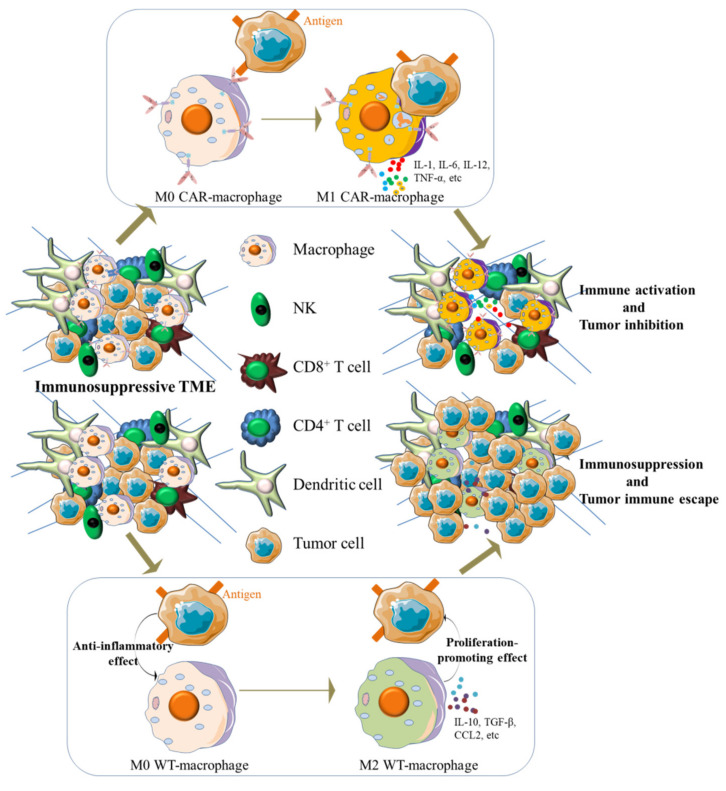
CAR-macrophages exert antigen-dependent cytotoxic activity against antigen-expressing tumor cells in vivo and contribute to the change of TME. When exposed to the surface antigen of tumor cells, M0 state CAR-macrophages switched into a pro-inflammatory phenotype (M1) and exerted an antitumor effect. Meanwhile, activated CAR-macrophages reprogramed the TME by releasing pro-inflammatory cytokines that can activate innate immune cells in TME, such as exhausted CD8^+^ T cells, while adoptive transplanted WT-macrophages will be induced into an immunosuppressive state (M2) and therefore orchestrated an immunosuppressive microenvironment.

## Data Availability

Not applicable.

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
