# Peer review of "Induced CAR-Macrophages as a Novel Therapeutic Cell Type for Cancer Immune Cell Therapies"

_cells, 2022, doi:10.3390/cells11101652_

Round 1
Reviewer 1 Report
I am satisfied with the corrections made and recommend the manuscript for publication.
Author Response
Thanks for your approval of this manuscript. To offer a better understanding, we further revised the manuscript. It would be our pleasure if the new vision could be approved by you.
Reviewer 2 Report
I have read the revised manuscript and found that the authors have addressed my major comments
I found the revised manuscript acceptable for publication
Please check references since some seem not correctly reported.
Author Response
Thanks for your precious comments and approval of this manuscript. We have checked all the reference carefully and revised references that might not be quite appropriate.
Reviewer 3 Report
The Review by Su and colleagues points to give an overview on the state of the art of CAR-Macrophages development. This topic is up-to-date and also scientifically very interesting and deserves to be deepened. Indeed, the application of adoptive immunotherapy with CAR-T cells in solid tumors still presents a challenge, and alternatives are urgently searched.
The manuscript is difficult to read due to the lack of a well-organized structure already in the introduction section, which should provide the readers with all the basic information necessary to a good understanding of the matter discussed. The AAs introduce CAR-Macrophages and try to highlight their potential importance as adoptive immunotherapy for solid cancer. However, the description of macrophages, their plasticity, and the functions assumed within the TME, is confusing and does not provide readers with a clear background.
Moreover, also the part of the MS which focuses on the source of human macrophages potentially suitable for the production of CAR-macrophages must be amply reviewed. I suggest AAs to give a summary on chimeric antigen receptor development history, and why a certain type has been chosen for CAR-macrophages. Although to date the literature on CAR-macrophages is limited because matter of study of the last couple of years, this point is particularly crucial and needs to be discussed.
Overall the MS does not provide new insights on CAR-macrophages topic compared to other recently published papers.
In my opinion, the manuscript cannot be considered for publication in the Journal “Cells” whose IF is of great value.
Author Response
Comment 1: “The manuscript is difficult to read due to the lack of a well-organized structure already in the introduction section, which should provide the readers with all the basic information necessary to a good understanding of the matter discussed. The AAs introduce CAR-Macrophages and try to highlight their potential importance as adoptive immunotherapy for solid cancer. However, the description of macrophages, their plasticity, and the functions assumed within the TME, is confusing and does not provide readers with a clear background.”
Answer:Thanks for your professional opinions and pertinent evaluations. The unfortunate paragraph structure does result in difficulty to read and understand. In organizing the “introduction” section, we originally intended to briefly introduce the concept of the CAR, and the successful application of CAR-T in the clinical treatment of hematologic malignancies and its current situation in the treatment of solid cancers firstly. We then focused on introducing the prospect of macrophages in the therapy of solid cancers by listing inherent features of the immune cells and giving an outline of influence on the progression of solid cancers. Importantly, we mainly emphasized this influence of macrophages from the aspects of recruiting Tregs and MDSCs, secreting inhibitory cytokines, inducing hyper-expression of PD-L1, and altering the metabolic profile of immune cells. Finally, we showed the different solid tumor therapies utilizing macrophages and their drawbacks. Through these series of logical introductions, we roused the application of adaptive transplantation of CAR-macrophages. In order to well understand the background, we have redrawn the paragraphs. We hope the revised versions can provide readers with a clear understanding of the matter discussed.
Comment 2: “Moreover, also the part of the MS which focuses on the source of human macrophages potentially suitable for the production of CAR-macrophages must be amply reviewed. I suggest AAs to give a summary on chimeric antigen receptor development history, and why a certain type has been chosen for CAR-macrophages. Although to date the literature on CAR-macrophages is limited because matter of study of the last couple of years, this point is particularly crucial and needs to be discussed. ”
Answer: These are very professional suggestions. Firstly, it would certainly improve the quality of this paper and offer more information about the application of macrophages, if we reviewed the source of human macrophages potentially suitable for the production of CAR-macrophages. According to the findings we can reach, tissue-resident macrophages are mainly originated from yolk sacs in the development of an organism. While the macrophages patrolling the blood vessel are differentiated from bone marrow-derived hematopoietic stem cells (HSC). However, just as you had pointed out that the limited literature on CAR-macrophages, we had not found the research that comparing which sources of macrophage are suitable for the production of CAR-macrophages. In fact, we found it is difficult to genetically modify all the innate macrophages in our research. Nevertheless, with the maturation of differentiation techniques, iPSC that possess genetic manipulation-friendly features have been regarded to be an ideal source of macrophage. We have pointed out this in this manuscript.
Secondly, in view of the limitation of the topic and numerous reports about CAR development, we focused the contents on research of therapy-using of macrophages for solid cancers in this paper, and abdicated the introduction of the development history of CAR. Meanwhile, in this manuscript, we briefly summarized the reason why the intracellular CD3ζ domain has been chosen for CAR-macrophages is the fact that the CD3ζ intracellular domain is homologous to Fc common γ-chain (FcεRI-γ), which is a specific classical signaling molecule in macrophages participating in antibody-dependent cellular phagocytosis (ADCP).
Reviewer 4 Report
The current review is well directed and provides an important and comprehensive update of the use of CAR-Mac /CAR-iMAC and provides sufficient introduction of CAR-T cell's use and its possible limitations in the treatment of solid tumors. In conclusion, the authors also suggested the possible advancements and limitations of the use of CAR-mac use in immunotherapy. The graphics provided in the manuscript is also well-curated. In the introduction, authors should describe the acronyms at their first use i.e. Tregs, MDSCs etc.
Author Response
Thanks for your kind and careful reminder. We have added the full names of the acronyms at their first use.
Round 2
Reviewer 3 Report
Although I appreciated the efforts spent by the AAs in revising the MS, especially concerning the revision of the Figures which are now really appealing, I still think that the MS still lacks a consistent structure. The matter of the study is new and interesting but needs to be adequately discussed.
As already pointed out in my previous report, a review article, irrespective of the complexity of the matter discussed, must be organized to give a coherent background and then developed taking into consideration the taking-home messages of the article.
If I properly understand, the AAs intend to compare the different sources of macrophages that can be used for the production of CAR-macrophages and also discuss their application in the efficient killing of cancer cells.
With this aim in mind, the introduction section needs to be organized. In the reply to my report, the AAs explained how they organized the introduction section. I think that their outline can be accepted, but all the points need to be well introduced and discussed and not just mentioned.
The structure of the introduction section has not changed at all, as I requested.
Overall the MS is difficult to read, also because it needs to be amply revised for the English language.
I still think that the MS cannot be considered for publication in this Journal.
Author Response
We really appreciate your valuable advice and agree with your professional comments. Due to the fact that the research of tumor immune cell therapy based on CAR-macrophages is just beginning and the limited information available in this field, we mainly focused on the introduction of current research progress of CAR-macrophages and made some simple discussions in this manuscript. More in-depth discussions may need to be supported by further research. Given your valuable advice, we will keep track of the latest research developments in this field and intend to present much more useful information to peers in near future.
Besides, to make the manuscript easy to understand, we have improved the English language with the help of a native English-speaking colleague. Besides, given the introduction of the influence of macrophages on the tumor microenvironment would confuse readers, we have moved this content to a new section.
Finally, we have improved the English language of this manuscript with the help of a native English-speaking colleague.
We hope these revisions and efforts could meet your requirement.
This manuscript is a resubmission of an earlier submission. The following is a list of the peer review reports and author responses from that submission.
Round 1
Reviewer 1 Report
Dear authors:
The topic of the review is very interesting, and I think many researchers working in the field of macrophages and tumours would read the review devoted to this topic with interest.
There are several fundamental remarks and critical comments.
- At the beginning of the section “1. Introduction”, first sentence “Macrophages have been defined as highly plastic cells,… .. [1]. ” contradicts the second sentence “Moreover, it has been reported that macrophages stay in two polarized states, i.e., classically activated (or inflammatory) macrophages (M1) and the other alternatively activated (or wound healing) macrophages (M2) [2]”.
Ref. 2 refers to a work published ten years ago. Modern concepts say that M1 and M2 are extreme polar states that can be achieved only under cell culture conditions. In a real situation, the macrophage phenotype can be in any state within the continuum between the M1 pro-inflammatory and M2 anti-inflammatory states.
- The authors declare "Induced CAR-Macrophages as a Novel Therapeutic Cell Type for Cancer Immune Cell Therapies." The review would greatly benefit if the authors introduced a separate section and analyzed and compared CAR-Macrophages with CAR-T cells. This comment can be taken as a wish to the authors.
- Line 182: "The anti-cancer movement of CAR-Ms was examined in NSGS mice [31]."
Ref. 31 "Wunderlich M, Chou FS, Link KA, et al. AML xenograft efficiency is significantly improved in NOD / SCID-IL2RG mice constitutively expressing human SCF, GM-CSF and IL-3. Leukemia. 2010; 24 (10): 1785-1788." is incorrect. There is no mention of CAR-Ms. in this 2010 article at all. This situation with citations is unacceptable. Authors should carefully check the correctness of the citation throughout the text.
- There is an impression of some sloppiness and haste in the formatting of the text: there are mistakes, misprints and quotations without proper formatting
Minor comments:
Line 51-53: In the sentence “Recent studies showed a new type of immune cells which are genetically engineered human macrophages with CARs, and therefore provided a novel choice for immunotherapy of neoplasms.” The authors should insert references to “recent studies.”
It is necessary to carefully review the text for misprints and spelling errors, such as in the sentence “Next, CAR-Ms Were validated phenotypic plasticity in vitro.” (Line 159) or "adaptive transplantation" (Line 246)
Please include the reference number, not just the author's name, for example, "Klichinsky et al." (Line 247) or "Klichinsky et al., Zhang et al." (Line 252). Similar omissions should be reviewed throughout the text.
An unusually large number of authors - 9 people! It may be advised to remove researchers who did not contribute or made minor contributions to the work.
Figure 1B is almost entirely redrawn, with minor changes, from Figure 1a from Zhang L, Tian L, Dai X, et al. Pluripotent stem cell-derived CAR-macrophage cells with antigen-dependent anti-cancer cell 317 functions. J Hematol Oncol. 2020; 13 (1): 153. In such cases, it is necessary to provide a link where the picture was taken from and the authors' permission. If minor changes are made, you need to provide a link from where to get the original drawing and write that the drawing has been slightly changed. Figure 1B cannot be left as it is. If the situation is similar to other Figures, it must also be corrected.
Reviewer 2 Report
In this manuscript, Su et al. propose a review of the recent literature in the emerging field of CAR-expressing macrophage cells (CAR-Mac), which is a promising new application of the chimeric antigen receptor (CAR)-T cell therapy. The review is sufficiently well organized, providing a useful introductory reading to CAR-Mac studies. However, the most relevant section of this review, which focuses on CAR-Mac in vitro and in vivo studies, is essentially based on the discussion of 2 manuscripts. This represents a major weakness of the current version of the review, which in the present version did not provide a comprehensive and updated revision of pre-clinical and clinical studies available on CAR-Mac.
Major Criticisms
- Limited discussion of CAR-Mac literature.
- References are missing or mislabeled (e.g. no references are included in Section 3.1.2 row 182-197; Ref. 30 does not correspond to the description reported in the main text)
Reviewer 3 Report
The paper is an interesting review about a novel and relevant topic. i.e. the possibility to modulate macrophage activation to incluence the immune response against tumors and the T cell activity. However, the paper need some revisions to be more complete and exhaustive.
Major comments:
Introduction, lines 30-31: please revise the order of function first immune actions against pathogens and consequently of their activation tissue homeostasis and changes
Introduction, line 40: please cite some relevant recent studies that show the better prognostic role of M1 macrophages in patients (not in vitro or in vivo). For example see Sci Rep. 2020 Apr 8;10(1):6096
Introduction, lines 42-43. As regard these concepts that are very relevant in the clinical practice and for the understanding of the efficacy of immunotherapy, the authors should explain and clarify the fundamental evidence that macrophages may contribute to immunosuppressive microenvironment and to T cell exhaustion by expressing PD1 ligand and by develop specific metabolic changes that influence T cell functions, metanolism and efficiency. At this regard the different cytokines and metabolism of Macrophages may drive different T cells populations. Please comment about this point.
Introduction, lines 51-52: please add the references of these studies.
Reviewer 4 Report
In principle, the authors present a interesting -but rather short- manuscript regarding the use of macrophages as CAR-effector cells.
Due to the rather low number of references this is probably more a mini-review than an full-blown review. In any case there are too many authors for the volume of the manuscript.
Contextual comments:
line 167-180: Does this paragrph refer to the aforementioned reference #30?
line 224: The text referes to reference #30 as Zhang et al., but #30 in the reference list ist not Zhang et al. Double-check references.
Figure 1: The CAR macrophages in panel A - and not only the iMACs in panel B - will also express CARs on their surface.
Figure 2: In this figure the relevant domains of CD3-Zeta containing CARs and CD19-M-CAR are shown, but their respective functions are not described in the caption or text. Especially for the understanding of readers who are not so familiar with the matter it would be useful to describe the functions of the individual domains.
In addition, I did not find a "competing interests" statment in the present paper indicating that Jin Zhang is the founder and scientific advisor of CellOrigin